# Single-cell chromatin accessibility profiling of glioblastoma identifies an invasive cancer stem cell population associated with lower survival

Paul Guilhamon[1,2], Charles Chesnelong[2], Michelle M Kushida[2], Ana Nikolic[3,4,5], Divya Singhal[3,4,5], Graham MacLeod[6], Seyed Ali Madani Tonekaboni[1,7], Florence MG Cavalli[2], Christopher Arlidge[1], Nishani Rajakulendran[6], Naghmeh Rastegar[2], Xiaoguang Hao[3,8,9], Rozina Hassam[3,8,9], Laura J Smith[10], Heather Whetstone[2], Fiona J Coutinho[2], Bettina Nadorp[1], Katrina I Ellestad[3,4], H Artee Luchman[3,8,9], Jennifer Ai-wen Chan[3,4,11], Molly S Shoichet[10], Michael D Taylor[2,12,13], Benjamin Haibe-Kains[1,7,14,15,16], Samuel Weiss[3,8,9,17], Stephane Angers[6,18], Marco Gallo[3,4,5,17], Peter B Dirks[2,12,15]*, Mathieu Lupien[1,7]*

[1]Princess Margaret Cancer Centre, University Health Network, Toronto, Canada; [2]Developmental and Stem Cell Biology Program and Arthur and Sonia Labatt Brain tumor Research Centre, The Hospital for Sick Children, Toronto, Canada; [3]Clark Smith Brain Tumour Centre, Arnie Charbonneau Cancer Institute, University of Calgary, Calgary, Canada; [4]Alberta Children's Hospital Research Institute, University of Calgary, Calgary, Canada; [5]Department of Biochemistry and Molecular Biology, Cumming School of Medicine, University of Calgary, Calgary, Canada; [6]Leslie Dan Faculty of Pharmacy, University of Toronto, Toronto, Canada; [7]Department of Medical Biophysics, University of Toronto, Toronto, Canada; [8]Hotchkiss Brain Institute, University of Calgary, Calgary, Canada; [9]Department of Cell Biology & Anatomy, University of Calgary, Calgary, Canada; [10]Institute of Biomaterials and Biomedical Engineering, University of Toronto, Toronto, Canada; [11]Department of Pathology and Laboratory Medicine, University of Calgary, Calgary, Canada; [12]Division of Neurosurgery, University of Toronto, Toronto, Canada; [13]Departments of Molecular Genetics and Surgery, University of Toronto, Toronto, Canada; [14]Department of Computer Science, University of Toronto, Toronto, Canada; [15]Ontario Institute for Cancer Research, Toronto, Canada; [16]Vector Institute, Toronto, Canada; [17]Department of Physiology & Pharmacology, University of Calgary, Calgary, Canada; [18]Department of Biochemistry, Faculty of Medicine, University of Toronto, Toronto, Canada

*For correspondence:
peter.dirks@sickkids.ca (PBD);
mlupien@uhnres.utoronto.ca (ML)

Competing interests: The authors declare that no competing interests exist.

**Abstract** Chromatin accessibility discriminates stem from mature cell populations, enabling the identification of primitive stem-like cells in primary tumors, such as glioblastoma (GBM) where self-renewing cells driving cancer progression and recurrence are prime targets for therapeutic intervention. We show, using single-cell chromatin accessibility, that primary human GBMs harbor a heterogeneous self-renewing population whose diversity is captured in patient-derived glioblastoma stem cells (GSCs). In-depth characterization of chromatin accessibility in GSCs identifies three GSC states: Reactive, Constructive, and Invasive, each governed by uniquely essential transcription factors and present within GBMs in varying proportions. Orthotopic xenografts reveal that GSC states associate with survival, and identify an invasive GSC signature

predictive of low patient survival, in line with the higher invasive properties of Invasive state GSCs compared to Reactive and Constructive GSCs as shown by in vitro and in vivo assays. Our chromatin-driven characterization of GSC states improves prognostic precision and identifies dependencies to guide combination therapies.

## Introduction

Glioblastoma (GBM) is a lethal form of brain cancer with standard surgery and radiation giving a median survival of only 12.6 months (*Carlsson et al., 2014*). The addition of temozolomide chemotherapy provides only an additional 2.5 months in the small subset of responsive patients (*Stupp et al., 2005*). Despite extensive characterization and stratification of the bulk primary tumors, no targeted therapies have been successfully developed (*Carlsson et al., 2014*; *von Neubeck et al., 2015*). GBM tumors are rooted in self-renewing tumor-initiating cells commonly referred to as glioblastoma stem cells (GSCs) (*Venere et al., 2011*) that drive disease progression in vivo (*Chen et al., 2012*; *Gallo et al., 2015*) and display resistance to chemo- and radiotherapy leading to disease recurrence (*Bao et al., 2006*). The promise of therapeutically targeting self-renewing tumor-initiating cancer cells depends on our capacity to capture the full range of heterogeneity within this population from individual tumors. Intratumoral heterogeneity within primary GBM has recently been documented through single-cell RNA-seq experiments and revealed a continuum between four cellular states (*Neftel et al., 2019*): neural-progenitor-like (NPC), oligodendrocyte-progenitor-like (OPC), astrocyte-like (AC), and mesenchymal-like (MES) (*Neftel et al., 2019*). A subsequent study (*Wang et al., 2019*) using single-cell gene-centric enrichment analysis placed GBM cells along a single axis of variation from proneural to mesenchymal transcriptional profiles, with cells expressing stem-associated genes lying at the extremes of this axis. Hence, primary GBM consists of distinct states, across which stem-like cells appear to be found. Whether these stem-like cells found across GBM states represent functionally distinct GSC populations with tumor-initiating properties and unique dependencies remains to be established to guide therapeutic progress. To address this issue, we combined single-cell technologies to define GSC composition in primary GBM with functional assays to reveal the unique dependencies across GSCs, reflective of invasive, constructive, and reactive states that relate to patient outcome.

## Results

Chromatin accessibility readily discriminates stem from mature cell populations (*Stergachis et al., 2013*), which can be resolved at the single-cell level through single-cell ATAC-seq (*Buenrostro et al., 2018*; *Corces et al., 2016*), taking into account non-gene centric features, such as accessibility of noncoding elements and total amount of accessible DNA sequences. Applying single-cell chromatin accessibility profiling (scATAC-seq) across four primary adult GBM tumors (3797 cells), wild type for both *IDH1* and *IDH2*, revealed seven to nine accessibility modules in each tumor based on unsupervised clustering (*Figure 1A*). We assigned cells to each of the four scRNA-seq-derived cellular states (*Neftel et al., 2019*) based on individual cells' chromatin accessibility enrichment scores for the promoter regions of each state's signature genes. Across the four tumors, 35–55.2% of the cells were significantly enriched for at least one state's signature genes (*Figure 1—figure supplement 1A*). The MES state reported from scRNA-seq (*Neftel et al., 2019*) dominates the identity of two or more modules reported from chromatin accessibility in every tumor (*Figure 1B,C*, *Figure 1—figure supplement 1B*). In contrast, the NPC and OPC states are mixed within the same module defined based on chromatin accessibility, dominating over the other states typically in at least two modules. Cells assigned to the AC state did not preferentially cluster within a single module reported from chromatin accessibility (*Figure 1B,C*, *Figure 1—figure supplement 1B*). Collectively, our results suggest that chromatin accessibility reflects a greater stratification of the MES state and detects similarities between the OPC and NPC states and heterogeneity within the AC state.

To identify putative cancer stem cells within each primary tumor, we next focused on the level of chromatin accessibility at promoters of 19 transcription factors previously associated with self-renewal and tumor-propagating capacity in GBM (*Suvà et al., 2014*; *Figure 1D*). Individual cells

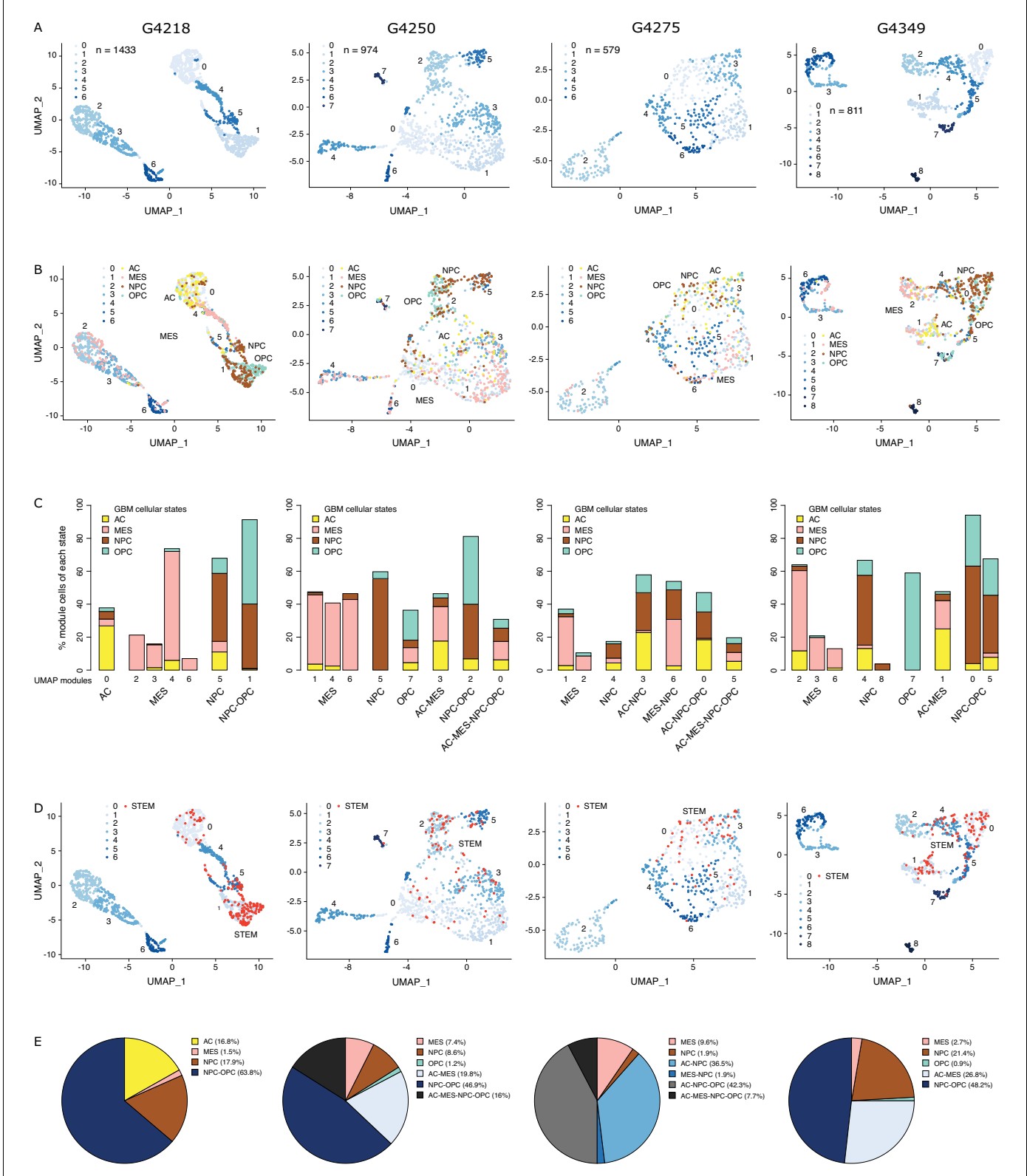

**Figure 1.** The diverse glioblastoma (GBM) cancer stem cell pool. (A) UMAP (Uniform Manifold Approximation and Projection) representation of chromatin accessibility across four primary GBM. (B) UMAPs with tumor cells assigned to cellular states. (C) UMAP modules are grouped by dominant cellular state. (D) UMAPs with cancer stem cells highlighted based on the enrichment of GBM cancer stem transcription factor promoters. (E) Distribution of cancer stem cells across the modules dominated by each cellular state.

*Figure 1 continued on next page*

*Figure 1 continued*

The online version of this article includes the following figure supplement(s) for figure 1:

**Figure supplement 1.** The distribution of cell states across tumor cells.

scoring as putative cancer stem cells were not restricted to a unique module defined by chromatin accessibility but were distributed across a subset of modules, suggesting heterogeneity across cancer stem cells in primary GBM, in agreement with reports relying on single GBM cell labeling (*Corces et al., 2016*; *Gallo et al., 2015*; *Lan et al., 2017*; *Liau et al., 2017*; *Meyer et al., 2015*; *Miller et al., 2017*; *Orzan et al., 2017*; *Park et al., 2017*; *Rheinbay et al., 2013*; *Stergachis et al., 2013*; *Suvà et al., 2014*), assessing the heterogeneity of self-renewing tumor-initiating cells. Putative cancer stem cells identified in primary GBM through scATAC-seq were found in modules ascribed to every one of the four cellular states defined by gene expression (*Neftel et al., 2019*), predominantly within NPC and OPC containing modules and a smaller fraction (<10%) in MES-specific modules across all four tumors (*Figure 1E*). This suggests that the core transcriptional unit of cancer stem cells in primary GBM (*Suvà et al., 2014*) is not restricted to a unique population defined by its global transcriptional or chromatin accessibility profile with the resolution achieved with current single-cell technologies.

To further probe the heterogeneity in chromatin accessibility within the GBM cancer stem cell pool, we derived GSC populations from 27 adult *IDH* wild-type GBM tumors (*Pollard et al., 2009*) and profiled their chromatin accessibility by bulk ATAC-seq (*Figure 2A*). Each patient-derived GSC showed a similar enrichment for accessible chromatin regions in promoters and 5'UTRs, and depletion in introns and distal intergenic regions (*Figure 2B*). Collectively, we uncovered 92% of the total predicted regions of accessible chromatin (255,891 regions) within GSCs based on a saturation analysis using a self-starting non-linear regression model across the 27 samples (*Figure 2C*). We next assessed the similarity between these GSCs and the putative cancer stem cells found by scATAC-seq in the four primary GBMs. GSCs were identified within each tumor by calculating the enrichment of accessible chromatin regions shared by a majority of GSCs (>14/27) in each tumor cell (*Figure 2D*). On average, 11.3% of cells in each primary GBM were labeled as GSCs. Comparing the distribution of GSCs across the seven to nine modules defined by scATAC-seq to that of the 19 transcription factor-derived cancer stem cell signature demonstrates concordance between the two signatures (*Figure 2E*). Moreover, the enrichment z-scores for both cancer stem signatures (i.e. stem transcription factors signature and GSC chromatin accessibility signature) are significantly correlated across cells in all four tumors (p$\leq$1.6$\times$10$^{-5}$) (*Figure 2F*). The overlap in cells significantly enriched for either stem signatures across the four tumors is also significant (hypergeometric test p-value=8.8$\times$10$^{-8}$) Additionally, an average of 91.2% (85.1–100%) of the cells identified by either signature display the hallmark GBM copy number changes at chromosomes 7 and 10, confirming their neoplastic status (*Figure 2—figure supplement 1*). Collectively, these results demonstrate that the patient-derived GSC populations reflect the chromatin identity of putative cancer stem cells found in primary brain tumors, highlighting the value of these GSCs as models to deepen our understanding of individual cells within primary GBM with features found in self-renewing tumor-initiating cells. Accordingly, spectral clustering of the 27 patient-derived GSC ATAC profiles identifies three distinct states of self-renewing tumor-initiating cells (*Figure 3A*). Expression profiling of these 27 GSCs by RNA-seq reveals GSCs significantly enriched for the signatures of each of the three TCGA GBM subtypes (proneural, classical, and mesenchymal) (*Wang et al., 2017*; *Figure 3B*). However, the assignment of the proneural, classical, and mesenchymal subtypes across GSCs did not match the three clusters identified from ATAC-seq (*Figure 3B*). Conversely, clustering the GSCs by gene expression, independently of their chromatin accessibility, did largely recapitulate the GSC states defined by chromatin accessibility (*Figure 3B*). This suggests that the mismatch between the GSC cluster from chromatin accessibility profiles and TCGA expression subtypes is not mainly due to differences between chromatin accessibility and gene expression. Potential alternative causes for this observed mismatch include the absence of a tumor microenvironment in the GSC populations or that given the TCGA subtypes were determined from bulk GBM, they may not fully capture the nature of rarer populations found within a tumor, such as the cancer stem cell populations.

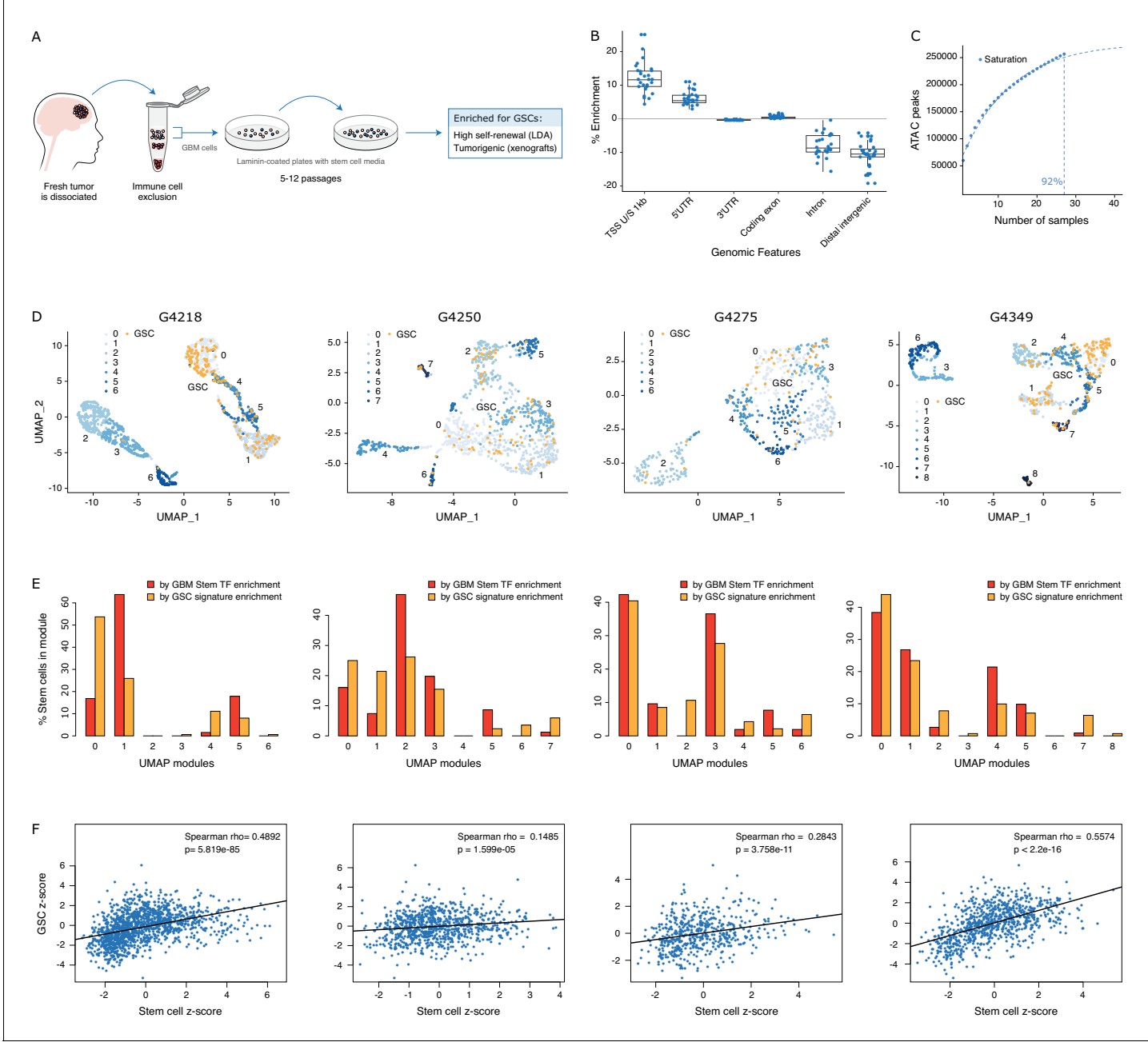

**Figure 2.** GSCs recapitulate the glioblastoma (GBM) cancer stem cell population. (A) Schematic representation of the GSC derivation process, from patient tumor to GSC-enriched population. (B) Genomic feature enrichment of accessible chromatin peaks. (C) Saturation curve for the 27 GSCs. (D) UMAPs with GSCs highlighted based on the enrichment of shared accessible regions across GSCs. (E) Proportion of UMAP modules assigned to cancer stem cells and GSCs. (F) Correlation of z-scores for each signature for each cell in each primary GBM.

The online version of this article includes the following figure supplement(s) for figure 2:

**Figure supplement 1.** Identification of tumor cells through characteristic copy number changes.

Gene set enrichment analysis with GSEA (Gene Set Enrichment Analysis) (*Subramanian et al., 2005*) and g:profiler (*Reimand et al., 2016*) using genes exclusively enriched for both expression and promoter chromatin accessibility in each subtype reported significantly enriched terms defining the largest GSC state as a Reactive state, with terms related to immune cells and response (*Figure 3C*, top panel). A second GSC state was enriched for Constructive gene sets involved in brain, neuron, and glial cell development (*Figure 3C*, middle panel). The third and smallest GSC

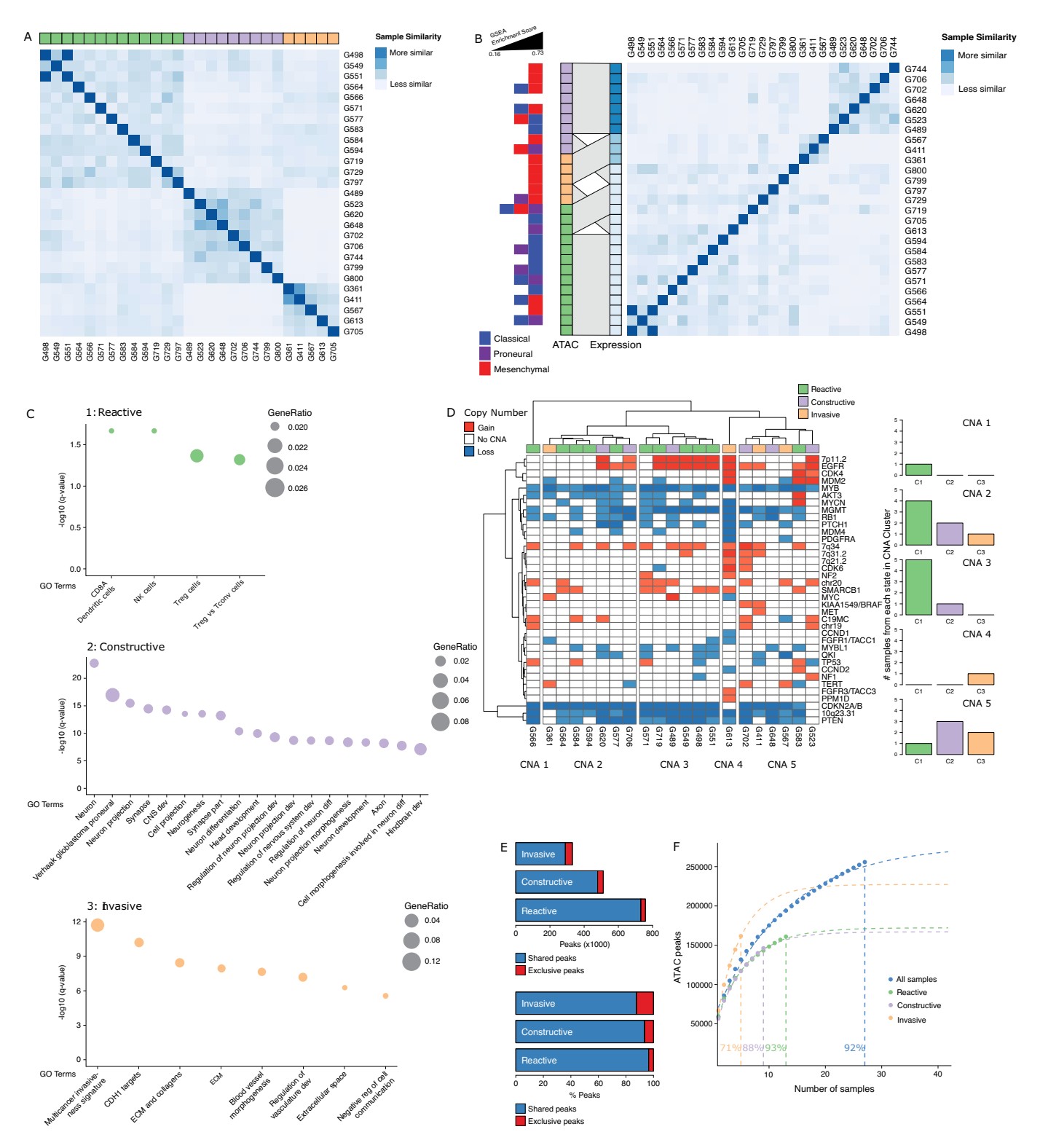

**Figure 3.** Three glioblastoma stem cell (GSC) states driven by chromatin accessibility. (**A**) Spectral clustering of ATAC-seq signal across peaks in 27 GSCs. (**B**) Spectral clustering of gene expression across GSCs and comparison to chromatin-derived GSC states. Enrichment of TCGA subtypes across GSCs and comparison to GSC states as determined by ATAC. The GSEA (Gene Set Enrichment Analysis) gene sets each contained 50 genes and the enrichment scores ranged from 0.16 to 0.73. (**C**) Gene set enrichment analysis in each GSC state. All displayed terms are significantly enriched (q-

*Figure 3 continued on next page*

*Figure 3 continued*

value < 0.05). (**D**) Copy number alterations (CNAs) across GSCs identified from DNA methylation array data cluster GSCs into five subgroups. (**E**) Number and percentage of peaks unique and shared in each GSC state. (**F**) Saturation analysis of each individual state.

The online version of this article includes the following figure supplement(s) for figure 3:

**Figure supplement 1.** Distinct chromatin accessibility profiles across glioblastoma stem cell states.

state presented an Invasive state characterized by terms relating to the extracellular matrix and angiogenesis (*Figure 3C*, bottom panel). We next mapped copy number alterations (CNAs) across the 27 GSCs by applying the molecular neuropathology classifier tool (*Capper et al., 2018*) to DNA methylation data from the GSCs (*Figure 3D*). While common GBM CNAs, including *EGFR* gains and *CDKN2A/B* loss, were observed, the CNA-based classification of GSCs failed to match the three chromatin accessibility-derived states, suggesting the three GSC states are not defined by somatic copy number events (*Figure 3D*). Further comparison of the accessible chromatin in each GSC state reveals that only a small subset of accessible chromatin regions drives the three GSC states (*Figure 3E*, *Figure 3—figure supplement 1*). Our ability to discriminate GSC state-specific regions of accessible chromatin is reflective of the comprehensiveness of our cohort to saturate the detection of accessible regions to 93%, 88%, and 71% across the Reactive (n = 13), Constructive (n = 9), and Invasive (n = 5) state GSCs, respectively (*Figure 3F*).

Considering that regions of accessible chromatin serve as binding sites for transcription factors engaging in gene expression regulation, we next tested for DNA recognition motif family enrichment across regions exclusively accessible in Reactive, Constructive, or Invasive GSC states (*Figure 4A*, *Figure 4—figure supplement 1A*). The most enriched DNA recognition motif families in each state were either depleted or showed low-level enrichment in the other states. Specifically, the DNA recognition motifs for the interferon-regulatory factor (IRF) and Cys2-His2 zinc finger (C2H2 ZF) transcription factor families were enriched in the Reactive state (*Figure 4A*, top panel). Regulatory factor X (RFX) and basic helix-loop-helix (bHLH) DNA recognition motifs were enriched in the Constructive state (*Figure 4A*, middle panel), while the Forkhead motif family was enriched in the Invasive state (*Figure 4A*, bottom panel). Genome-wide CRISPR/Cas9 essentiality screens (*Figure 4—figure supplement 1B*) in three Reactive, two Constructive and one Invasive GSC (*MacLeod et al., 2019*) revealed the preferential requirement for expressed transcription factors (*Figure 4—figure supplement 1C–E*) recognizing the enriched DNA recognition motif in a state-specific manner (*Figure 4B*). Specifically, the SP1 regulatory network is preferentially essential in the Reactive state GSCs (*Figure 4B*, top panel), *ASCL1*, *OLIG2*, *AHR*, and *NPAS3* are uniquely essential in the Constructive state GSCs (*Figure 4B*, middle panel) and *FOXD1* is essential only in the Invasive state GSC (*Figure 4B*, bottom panel). Notably, *SP1* itself is exclusively essential in only one Reactive GSC (G564). However, of the 36 transcription factors from the Reactive-enriched families (IRF and C2H2 ZF) that were essential in at least one Reactive GSC and not in any of the Constructive or Invasive GSCs, 13 are directly regulated by SP1 (*Figure 4—figure supplement 1C*), thus suggesting that the SP1 regulatory network as a whole, rather than SP1 on its own, is key in the Reactive GSC state. Notably, all six transcription factors display significantly higher expression in GBM compared to normal brain (*Tang et al., 2017*; *Figure 4—figure supplement 1F*), further supporting their function as key regulators of tumor initiation and development. An additional gene set enrichment analysis combining genes exclusively essential to each state with the putative targets of the key transcription factors outlined above identifies additional enriched terms supporting the identities of the three GSC states as Reactive, Constructive, and Invasive (*Figure 4—figure supplement 1G*).

Previous work suggests that GBM tumors harbor a heterogeneous population of GSCs (*Lan et al., 2017*; *Meyer et al., 2015*; *Patel et al., 2014*). We therefore quantified the presence of Reactive, Constructive, and Invasive cancer stem cells in our four primary GBM based on their scATAC-seq profiles. The Constructive state was dominant in every primary tumor, ranging from 9 to 21% of all cells captured by scATAC-seq (*Figure 4C*). The Reactive and Invasive states accounted for only 0–9% of all cells, in varying proportions from one tumor to another (*Figure 4C*). Collectively, our results further support the heterogeneous nature of cancer stem cells that populate primary tumors.

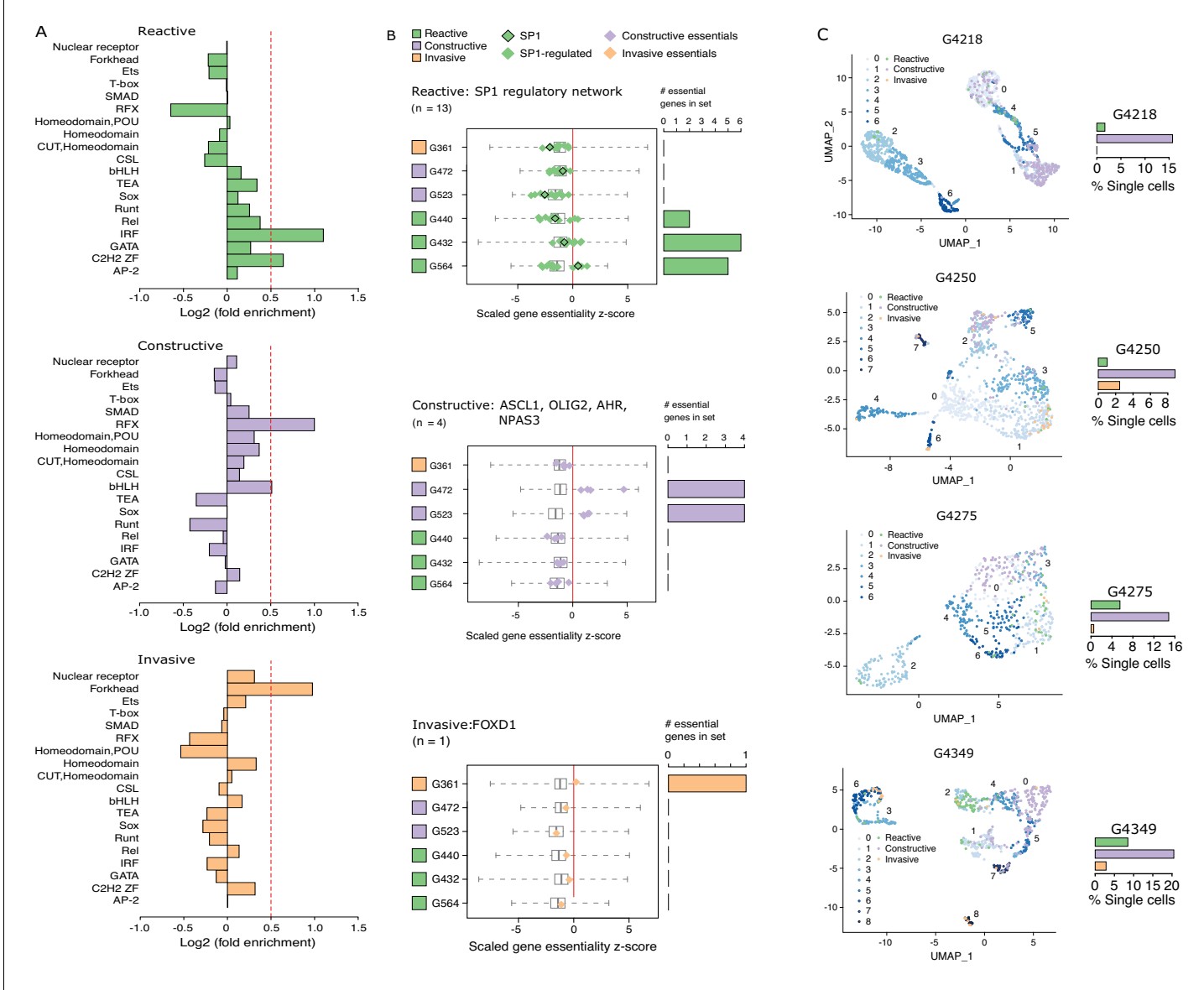

**Figure 4.** Functional diversity between glioblastoma stem cell (GSC) states and intra-tumor heterogeneity. (**A**) Motif family enrichment in each cluster; log2(fold enrichment) > 0.5 threshold selected based on the distribution of values in each cluster (*Figure 2—figure supplement 1*). (**B**) Z-score distribution of key essential genes in each cluster. Red line corresponds to the empirically determined threshold for essentiality in each tested line, scaled, and adjusted to zero. Boxplot whiskers in this case extend to data extremes. Side barplots show the total count of the key cluster-specific regulators found essential in each subtype. (**C**) UMAPs with GSCs from each state highlighted based on the enrichment of the top differentially accessible regions in each GSC state.

The online version of this article includes the following figure supplement(s) for figure 4:

**Figure supplement 1.** dentification and characterization of essential transcription factors across GSC states.

While various classifications of GBMs and/or their constitutive bulk and stem tumor cells have been reported, with some associating with patient survival (*Neftel et al., 2019*; *Patel et al., 2014*; *Cancer Genome Atlas Research Network et al., 2010*; *Wang et al., 2017*; *Yin et al., 2019*; *Zuo et al., 2019*), molecular signatures in adult GBM stratifying the poorer prognosis *IDH* wild-type patients by survival are lacking. We performed intracranial xenografts of 37 *IDH* wild-type GSC populations and classified the transplanted cells by their GSC state to perform a differential survival analysis (*Figure 5A*). The overall survival times of the transplanted mice grouped by GSC state were significantly different (LogRank test p=0.041), with the Invasive state GSCs leading to the worst

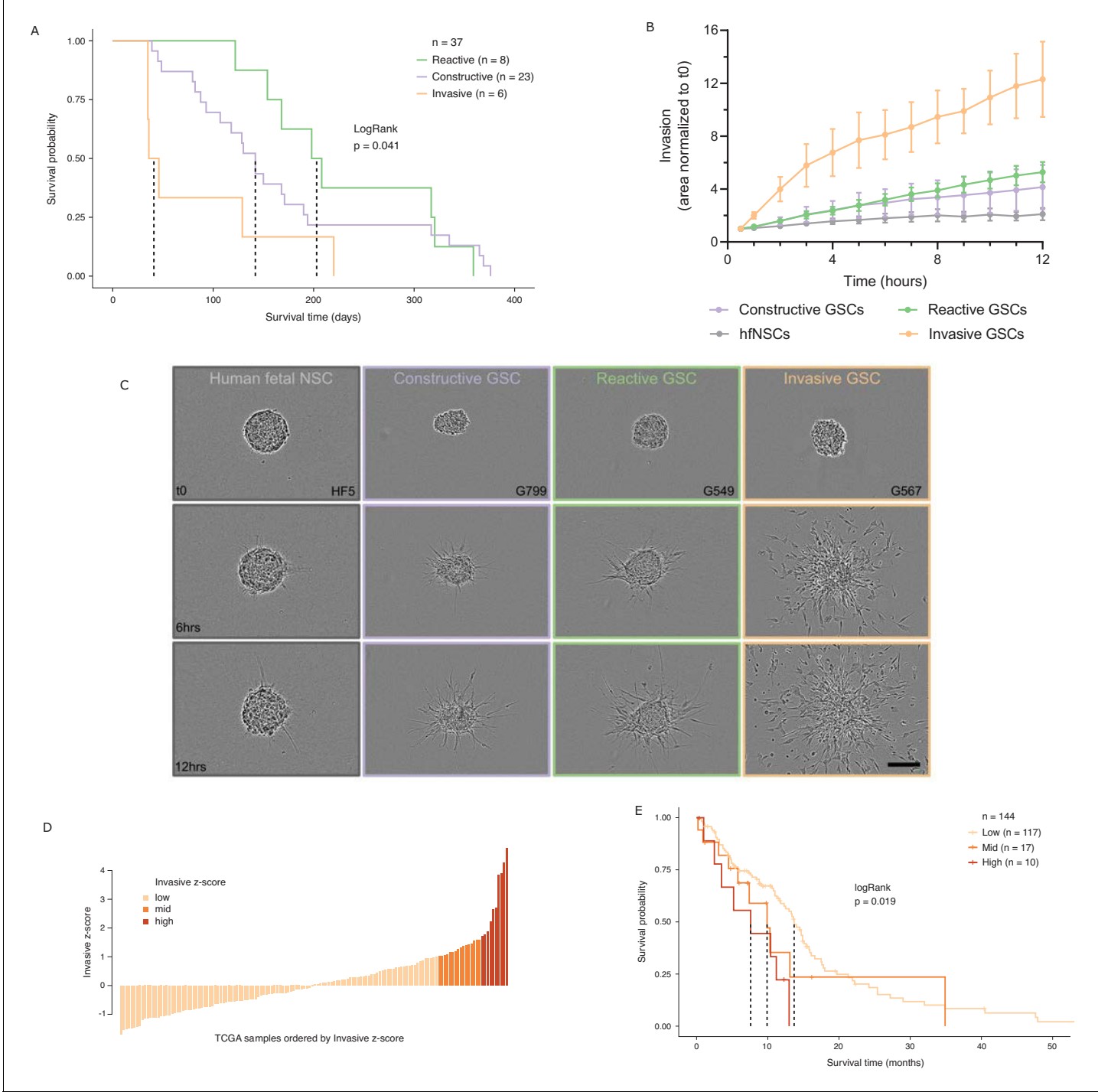

**Figure 5.** Glioblastoma stem cell (GSC) invasiveness is associated with survival in glioblastoma (GBM). (**A**) Kaplan–Meier plot for orthotopic xenografts grouped by GSC state. For each of the 37 GSCs used in the xenograft survival analysis, the median survival value was used from multiple mice injected with cells from each GSC (average number of mice injected/GSC = 5). The dotted lines indicate median survival. The pairwise p-values are also significant for Invasive vs Reactive (p=0.02) and Invasive vs Constructive (p=0.045) but not for Reactive vs Constructive (p=0.45). (**B**) Invaded area over time normalized to t0 by three to four representative GSCs of each GSC state and human fetal neural stem cells (hfNSCs) used as controls. See *Figure 5—figure supplement 1A* for individual GSC invasion. (**C**) Invasion images of representative hfNSCs and GSCs at t0, 6 hr, and 12 hr. Scale bar is 150 µm. (**D**) TCGA samples ordered by increasing concordance with Invasive GSCs and grouped into three subgroups: <1, 1–1.65, >1.65. (**E**) Kaplan–Meier plot for TCGA samples grouped by concordance with Invasive GSCs. The dotted line indicates median survival. When considering pairwise comparisons, only the Invasive-high and Invasive-low subgroups were significantly different (p=0.0043). Further subgrouping of the TCGA samples into

*Figure 5 continued*

smaller intervals of concordance z-score yielded no benefit, preserving the Invasive-high subgroup as the only one with significantly poorer prognosis (*Figure 5—figure supplement 2B,C*).

The online version of this article includes the following figure supplement(s) for figure 5:

**Figure supplement 1.** Invasive properties across glioblastome stem cell states.

**Figure supplement 2.** Assessment of Invasive GSC score classification.

prognosis. Relative to the mice injected with Reactive state GSCs, mice with Constructive state GSCs and Invasive state GSCs had hazard ratios of 1.3 (95% confidence interval [CI]: 0.57–2.97) and 3.5 (95% CI: 1.2–10.49), respectively.

In light of this finding, we sought to evaluate the invasive properties of the Invasive GSC state population. To that end, we conducted an invasion assay (*Restall et al., 2018*) wherein neurospheres from representative GSCs of each state (four Reactive, three Constructive, and three Invasive; three independent biological replicates for each GSC) are embedded in a collagen matrix and invasion is monitored in real time (*Figure 5B,C*, *Figure 5—figure supplement 1A*). GSCs of the Invasive state invaded strikingly faster and further into the collagen matrix and overall displayed a three times greater invasive ability than that of the Reactive and Constructive states. This invasiveness was further confirmed in vivo through human-specific staining of mouse brains injected with Invasive-state GSCs (*Figure 5—figure supplement 1B–C*).

Next, we investigated the prognostic value of the GSC states using the TCGA GBM cohort (*IDH1* and *IDH2* wild type, n = 144). When classified by dominant GSC state, TCGA tumors display the same trend as the xenografts with Invasive state-dominated tumors showing the lowest survival (*Figure 5—figure supplement 2A*). However, with only two tumors classified as Invasive-dominant, the difference in survival between the three patient groups was not statistically significant (p=0.3) (*Figure 5—figure supplement 2A*). We proceeded to rank the TCGA tumors solely by their concordance to Invasive GSCs and classified the patient tumors into Invasive-low (z-score < 1), Invasive-mid (z-score = 1–1.65), and Invasive-high (z-score $\geq$ 1.65) groups (*Figure 5D*). With this stratification method, median patient survival per group not only decreased with increasing Invasive GSC score, but we also identified an Invasive-high subset of tumors with significantly lower survival (p=0.019, HR (Hazard Ratio) = 2.8, 95% CI: [1.3–5.81]) (*Figure 5E*). This was further validated using an additional cohort of TCGA samples with microarray gene expression (*Figure 5—figure supplement 2D–E*). These results show that cancer stem cell states defined based on the chromatin accessibility in GSCs can identify transcriptional programs associated with poor prognosis and can serve as a signature to identify high-risk patients in *IDH* wild-type GBM.

## Discussion

Defining the nature of self-renewing tumor-initiating cells in primary GBM is required to identify vulnerabilities for therapeutic intervention. Quantifying their heterogeneity within tumors can guide treatment strategies and assist in predicting the course of disease progression. Here we show that chromatin accessibility assays capture a heterogeneity across self-renewing tumor-initiating cells in primary GBM that extends beyond their genetic diversity and underlies the heterogeneity in bulk progeny (*Meyer et al., 2015*). This heterogeneity aligns with diversity in the three-dimensional genome organization of GSCs (*Johnston et al., 2019*) and agrees with how the three-dimensional genome organization instructs cis-regulatory plexuses underlying gene regulation (*Bailey et al., 2016*; *Kim et al., 2016*; *Sallari et al., 2017*; *Schmitt et al., 2016*; *Zheng and Xie, 2019*). While the chromatin accessibility signature derived from GSCs robustly identified cancer stem cells across the four primary GBM tumors used in this study, a larger cohort will be required to establish whether this signature is always sufficient to capture all cancer stem cells in primary GBMs.

We further reveal a specific cancer stem state that is significantly predictive of patient survival and can be used as a signature to identify high-risk patients. Specifically, the Invasive GSC signature draws from a population with enhanced ability to invade and spread compared to other GSC states. Collectively, our results argue for distinct GSC populations whose composition in tumors impacts survival.

Our results also highlight dependencies unique to each cancer stem state. Specifically, the Reactive GSC state-associated transcription factor SP1 and its regulatory partners are involved in cellular differentiation and growth, apoptosis, response to DNA damage, chromatin remodeling (*O'Connor et al., 2016*), stimulation of *TERT* expression in cancer stem cells (*Liu et al., 2016*), and increased stemness and invasion in GBM (*Lee et al., 2014*). In contrast, the Constructive GSC state relies on transcription factors including OLIG2, a known GSC marker (*Trépant et al., 2015*); AHR involved in tumor microenvironment responses and metabolic adaptation (*Gabriely et al., 2017*); NPAS3, a regulator of Notch signaling and neurogenesis (*Michaelson et al., 2017*); and ASCL1, a critical regulator of GSC differentiation and marker of sensitivity to Notch inhibition in GSCs (*Park et al., 2017*; *Rajakulendran et al., 2019*). Finally, the Invasive GSC state relies on FOXD1, a pluripotency regulator and determinant of tumorigenicity in GSCs regulating expression of the aldehyde dehydrogenase ALDH1A3, a functional marker for invasive GSCs (*Cheng et al., 2016*; *Koga et al., 2014*). These results support developing combination therapy using targeting agents against each GSC state, such as Notch inhibitors (*Park et al., 2017*) and small molecule inhibitors of ALDH (*Cheng et al., 2016*), to eradicate self-renewing tumor-initiating cells with the hope to cure GBM patients.

# Materials and methods

## Patient samples and cell culture

All tissue samples were obtained following informed consent from patients, and all experimental procedures were performed in accordance with the Research Ethics Board at The Hospital for Sick Children (Toronto, Canada), the University of Calgary Ethics Review Board, and the Health Research Ethics Board of Alberta – Cancer Committee (HREBA). Approval to pathological data was obtained from the respective institutional review boards. Patient tumor tissue samples were dissociated in artificial cerebrospinal fluid followed by treatment with enzyme cocktail at 37°C. Patient tumor-derived GSCs were grown as adherent monolayer cultures in serum-free medium (SFM) as previously described (*Pollard et al., 2009*). Briefly, cells were grown adherently on culture plates coated with poly-L-ornithine and laminin. Serum-free NS cell self-renewal media (NS media) consisted of Neurocult NS-A Basal media, supplemented with 2 mmol/L L-glutamine, N2 and B27 supplements, 75 µg/mL bovine serum albumin, 10 ng/mL recombinant human EGF (rhEGF), 10 ng/mL basic fibroblast growth factor (bFGF), and 2 µg/mL heparin. A subset (22/37) of the GSCs used for orthotopic xenografts were grown as non-adherent spheres prior to single-cell dissociation and injection into the mice. Briefly, SFM was used to initiate GSC cultures. Non-adherent spheres formed after 7–21 days in culture and were expanded, then cryopreserved in 10% dimethyl sulfoxide (DMSO; Sigma-Aldrich) in SFM until used in experiments.

## ATAC-seq

ATAC-seq was used to profile the accessible chromatin landscape of 27 patient tumor-derived GSCs. Fifty thousand cells were processed from each sample as previously described (*Buenrostro et al., 2013*; *Corces et al., 2017*). The resulting libraries were sequenced with 50 bp single-end reads, which were mapped to hg19. Reads were filtered to remove duplicates, unmapped or poor-quality (Q < 30) reads, mitochondrial reads, chrY reads, and those overlapping the ENCODE blacklist. Following alignment, accessible chromatin regions/peaks were called using MACS2. Default parameters were used except for the following: `-keep-dup` all `-B -nomodel -SPMR` -q 0.05 `-slocal` 6250 `-llocal` 6250. The signal intensity was calculated as the fold enrichment of the signal per million reads in a sample over a modeled local background using the bdgcmp function in MACS2. Spectral clustering implemented in the SNFtool package (*Wang et al., 2014*) was run on the SNF-fused similarity matrix to obtain the groups corresponding to k = 2–12. Enrichment for genomic features was calculated using CEAS (*Shin et al., 2009*). The GSC state labels for each sample can be found in *Supplementary file 1*, and the coordinates of discriminating ATAC regions for each GSC state are in *Supplementary file 2*.

A hypergeometric test was used to determine whether there was any sex bias within the three GSC states. All tests resulted in p-value>0.05 except in the Invasive cluster where three out of five patients are female, giving a p-value=0.047. Given the small size of the Invasive group and the

impossibility of obtaining an even number of males and females in a group of 5, this result was not considered to indicate sex bias within the cohort.

A given chromatin region was considered exclusive to one of the clusters if it was called as a peak in any of the cluster's samples using a q-value filter of 0.05 and was not called as a peak in any of the other samples using a q-value filter of 0.2, in order to ensure stringency of exclusivity.

The ATAC-seq saturation analysis was performed by randomizing the order of samples and successively calculating the number of additional peaks discovered with the addition of each new sample. This process was repeated 10,000 times and averaged. A self-starting non-linear regression model was then fitted to the data to estimate the level of saturation reached.

For the xenograft survival analysis, 11/37 GSCs used overlap with the cohort of 27 described above. The other 26/37 GSCs were profiled by ATAC-seq independently following the same protocol described above and assigned to a GSC state as described below. Similarly, samples G432, G440, and G472 used in the essentiality gene analysis were profiled by ATAC-seq independently following the same protocol described above and assigned to a GSC state as follows: the signal obtained from MACS2 for each sample was mapped to the peak catalog of the original cohort of 27 GSCs. Each sample was then allocated to a GSC state through unsupervised hierarchical clustering with the original cohort of 27 GSCs.

## Single-cell ATAC-seq

The four tumors used were G4218 (primary GBM, *IDH* wt, male, 64 years), G4250 (primary GBM, *IDH* wt, male, 73 years), G4275 (primary GBM, *IDH* wt, female, 52 years), and G4349 (primary GBM, *IDH* wt, male, 62 years). Fragments of tumor were received fresh from the operating room, and blunt dissected into individual fragments of approximately 0.3–0.7 cm$^3$. Each fragment was placed in 1 mL of freezing media (400 µL of NeuroCult NS-A Basal medium with proliferation supplement (StemCell Technologies; #05751) containing 20 µg/mL rhEGF (Peprotech, AF-100–15), 10 µg/mL bFGF (StemCell Technologies, #78003), and 2 µg/mL heparin (StemCell Technologies, #07980); 500 µL of 25% bovine serum albumin (BSA) (Millipore-Sigma; A9647) in Dulbecco's modified Eagle's medium, and 100 µL DMSO (Millipore-Sigma; D2650) in a 2 mL cryotube, and placed at −80 C in a CoolCell for at least 24 hr. Samples were then stored at −80C until use. Cryopreserved primary GBM samples were washed at 1000 RPM for 5 min in phosphate-buffered saline (PBS) to remove DMSO, and then transferred to 1.5 mL tubes. Samples were resuspended in cold ATAC resuspension buffer (10 mM Tris–HCl pH 7.4, 10 mM NaCl, 3 mM MgCl$_2$, 0.1% NP-40, 0.1% Tween-20, 0.01% Digitonin, 1% BSA in PBS) on ice and dissociated using a wide-bore P1000 pipette tip and vortexing, followed by 10 min of incubation on ice. Cells were spun down at 500 x g for 5 min at 4°C, washed in the ATAC resuspension buffer, spun down again, and resuspended in ATAC-Tween wash buffer (10 mM Tris–HCl pH 7.4, 10 mM NaCl, 3 mM MgCl2, 0.1% Tween-20, 1% BSA in PBS), then passed through a cell strainer top FACS tube (Falcon; #38030) to remove debris. Nuclei quality and quantity was evaluated using trypan blue on an Invitrogen Countess II device in duplicate, and a subset of nuclei was spun down in a fresh tube and resuspended in 10× sample dilution buffer. Nuclei were then used for single-cell ATAC-seq library construction using the Chromium Single Cell ATAC Solution v1.0 kit (10× Genomics) on a Chromium controller. Completed libraries were further quality checked for fragment size and distribution using an Agilent TapeStation prior to sequencing. Single-cell ATAC-seq samples were sequenced on a NextSeq 500 (Illumina) instrument with 50 bp paired-end reads at the Centre for Health Genomics and Informatics (CHGI) at the University of Calgary.

The raw sequencing data was demultiplexed using cellranger-atac mkfastq (Cell Ranger ATAC, version 1.0.0, 10× Genomics). Single-cell ATAC-seq reads were aligned to the hg19 reference genome (hg19, version 1.1.0, 10× Genomics) and quantified using cellranger-atac count function with default parameters (Cell Ranger ATAC, version 1.1.0, 10× Genomics). The resulting data were analyzed using the chromVAR (*Schep et al., 2017*) and Signac (*Stuart et al., 2019*) R packages (v1.4.1). The number of accessibility modules in each sample was determined using the ElbowPlot method implemented in Signac. Similarity between individual cells and GSC states was assessed using the deviation scores calculated by chromVAR within the single-cell data for significantly differentially accessible sets of peaks (fold change signal difference > 2 and Wilcoxon test q-value ≤ 0.05) between the states as determined by bulk ATAC-seq. Similarity between individual cells and the

expression-derived cellular states was assessed using the deviation scores calculated by chromVAR within the single-cell data for promoter regions of the signature genes of each of the cellular states (*Neftel et al., 2019*). A twofold cut-off was used to determine the dominance of a UMAP module by an individual or group of cellular states. Similarity between individual cells and the GBM cancer stem cell signatures was assessed using the deviation scores calculated by chromVAR within the single-cell data for either promoter regions of the 19 transcription factors identified as markers of cancer stem cells in GBM (*Suvà et al., 2014*) or accessible chromatin regions shared by a majority of GSCs (>14/27). An average of 11.3% of cells in each tumor was identified as a GSC: G4218: 11.3%, G4250: 8.6%, G4275: 8.1%, G4349: 17.4%.

Copy number variants in single cells were determined using CONICSmat (*Müller et al., 2018*) with default parameters using the gene activity matrix generated by Signac as input. We focused on chr7 gains and chr10 losses as they are hallmark chromosomal changes in GBM and found the following fractions of cells carrying these CNVs, on average across the four tumors: 76% of all cells, 88% of cells allocated to scRNA-seq cellular states (*Neftel et al., 2019*), 95% of cancer stem cells based on the 19 gene signature, 91% of GSCs based on shared accessible regions between 14/27 GSC populations, and 94% of GSCs identified based on the state-specific signatures.

## DNA methylation arrays

Bisulfite conversion of DNA for methylation profiling was performed using the EZ DNA Methylation kit (Zymo Research) on 500 ng genomic DNA from all 27 samples. Conversion efficiency was quantitatively assessed by quantitative PCR (qPCR). The Illumina Infinium MethylationEPIC BeadChips were processed as per manufacturer's recommendations. The R package ChAMP v2.6.4 (*Morris et al., 2014*) was used to process and analyze the data. For the copy number analysis, the raw IDAT files were uploaded to the MNP tool (*Capper et al., 2018*), which directly compares the copy number profile estimated from the probe intensities on the methylation array to the distribution observed across thousands of brain tumors in its database.

## RNA-seq

RNA was extracted from GSCs using the Qiagen RNeasy Plus kit. RNA sample quality was measured by Qubit (Life Technologies) for concentration and by Agilent Bioanalyzer for RNA integrity. All samples had RIN above 9. Libraries were prepared using the TruSeq Stranded mRNA kit (Illumina). Two hundred nanograms from each sample were purified for polyA tail containing mRNA molecules using poly-T oligo attached magnetic beads, then fragmented post-purification. The cleaved RNA fragments were copied into first strand cDNA using reverse transcriptase and random primers. This is followed by second strand cDNA synthesis using RNase H and DNA Polymerase I. A single 'A' base was added and adapter ligated followed by purification and enrichment with PCR to create cDNA libraries. Final cDNA libraries were verified by the Agilent Bioanalyzer for size and concentration quantified by qPCR. All libraries were pooled to a final concentration of 1.8 nM, clustered, and sequenced on the Illumina NextSeq500 as a pair-end 75 cycle sequencing run using v2 reagents to achieve a minimum of ~40 million reads per sample. Reads were aligned to hg19 using the STAR aligner v2.4.2a (*Dobin et al., 2013*), and transcripts were quantified using RSEM v1.2.21 (*Li and Dewey, 2011*) or vst transformed using DESeq2 (*Anders and Huber, 2010*).

## Motif enrichment

Regions exclusively accessible in one of the GSC states and not the others were used as input sequences for the motif enrichment, while the full ATAC-seq catalog served as the background set when running HOMER v4.7 to detect enrichments of transcription factor binding motifs. Enriched motifs were then grouped into families based on similarities in DNA-binding domains using the CIS-BP database (*Weirauch et al., 2014*). Each family was assigned the fold-enrichment value of the most enriched motif within the family.

The transcription factors whose motifs were found enriched in Reactive-exclusive accessible regions were run together through GSEA (*Subramanian et al., 2005*), and the gene set corresponding to genes potentially regulated by SP1 was identified as significantly enriched (GSEA gene set

GGGCGGR_SP1_Q6). The expression levels of key transcription factors in tumor and normal samples were analyzed and displayed using GEPIA (*Tang et al., 2017*).

## Gene essentiality screen

Illumina sequencing reads from genome-wide TKOv1 CRISPR screens in patient-derived GSCs (*MacLeod et al., 2019*) were mapped using MAGECK (*Li et al., 2014*) and analyzed using the BAGEL algorithm with version two reference core essential genes/non-essential genes (*Hart et al., 2017*; *Hart et al., 2015*). Resultant raw Bayes factor (BF) statistics were used to determine essentiality of transcription factor genes using a minimum BF of 3 and a 5% false discovery rate cut-off. For visualization purposes only, the essentiality scores were scaled and the individual GSC essentiality thresholds subtracted from each score to obtain a common threshold at 0 across GSCs. In the GSEA analysis of essentiality genes, only those genes found essential only in a given GSC state were used, giving sets of 12, 29, and 235 genes for the Reactive, Constructive, and Invasive GSC states, respectively.

## Orthotopic xenografts

All animal procedures were performed according to and approved by the Animal Care Committee of the Hospital for Sick Children or the University of Calgary. All attempts are made to minimize the handling time during surgery and treatment so as not to unduly stress the animals. Animals are observed daily after surgery to ensure there are no unexpected complications. For intracranial xenografts, 100,000 GSC cells were stereotactically injected into the frontal cortex of 6–8 weeks old female NOD/SCID or C17/SCID mice. Mice were monitored and euthanized once neurological symptoms were observed or at the experimental end point of 12 months.

## Invasion assay

GSCs and human fetal neural stem cells (hfNSCs; used as controls) were seeded in flasks maintained vertically to limit adherence and incubated at 37°C, 5% $CO_2$ until the average neurosphere size reached approximately 150 µm. Neurospheres were then collected and allowed to settle by gravity to the bottom of a prechilled 1.5 mL conical tubes for 5 min on ice. Following aspiration of the supernatant, spheres were re-suspended in 0.4 mg/mL type I collagen (Cultrex Rat Collagen I, Trevigen, 3440–005) on ice. The suspended spheres were dispensed in prechilled 96-well plates (100 µL/well). The plates were maintained on ice for 5 min to allow spheres to settle at the same level at the bottom of the well and then transferred to the incubator at 37°C for 15 min to allow polymerization of the collagen. Plates were then placed in an IncuCyte (Essen Bioscience, Ann Arbor, MI), and invasion was monitored every hour for 12 hr. To quantify invasion of the cells from the embedded spheres into the collagen matrix, the area of the spheres at each time point was normalized to the area of the spheres at T0. Invasion experiments were performed at least in triplicates for each GSC and hfNSC lines. For each replicate, invasion was measured based on a minimum of three spheres.

## Immunohistochemistry

Tissue samples were formalin fixed and paraffin embedded. Serial sections deparaffinized and rehydrated through an alcohol gradient to water, and antigen retrieval in citrate buffer pH 6.0 was used for the human nucleolin antibody at 5.0 g/mL (ab13541) (Abcam, Cambridge, MA). Endogenous peroxide activity and nonspecific binding were blocked with 3%(vol/vol) peroxide and 2% (vol/vol) normal horse serum. Primary antibody and anti-mouse ImmPRESS-HRP secondary antibody were incubated for 1 hr and visualized using 3,3'-diaminobenzidine (Vectorlabs, Burlingame, CA). Normal horse serum or monoclonal IgM was used in control sections.

## Survival analysis

Survival analysis on xenografts and TCGA data was performed using R packages survival (*Therneau, 2015*) and survminer ('Drawing Survival Curves using 'ggplot2' [*R package survminer, 2020*]', n.d.). The LogRank test was used in every analysis. See ATAC-seq section for details on how each GSC used in the orthotopic xenografts was assigned to a GSC state. For each of the 37 GSCs used in the xenograft survival analysis, the median survival value was used

from multiple mice injected with each GSC. Each GSC was injected into one to six mice (mean number of mice used = 5; only one GSC has a single replicate: G489, a Constructive state GSC).

TCGA samples for this analysis were selected as follows: 144 IDH-wt adult GBM samples for which RNA-seq, IDH mutation status, and survival information were available. TCGA samples were assigned to individual GSC states in the following ways. (1) Using the unsupervised clustering of RNA-seq data presented in *Figure 3B*, the 23/27 GSCs that displayed matched GSC state assignments by RNA-seq and ATAC-seq were used in this analysis. (2) Genes preferentially enriched in each GSC state were determined using DEseq2 (*Anders and Huber, 2010*) (q $\leq$ 0.05 and fold change $\geq$ 2). (3) The mean log2(FPKM+1) value for each of these genes over all GSCs in each state was calculated to obtain a single representative value for each gene in each of the three GSC states. (4) The concordance index was then calculated between each TCGA sample and each GSC state, and individual TCGA samples were assigned to the GSC state with the highest score. Similarly, to assign TCGA samples to the three Invasive groups (Invasive-low, -mid, and -high), the concordance to Invasive GSCs as calculated above was used. The z-score for each sample was then used to classify each TCGA sample into the three subgroups of Invasive-low (Invasive z-score < 1), Invasive-mid (Invasive z-score = 1–1.65), and Invasive-high (Invasive z-score $\geq$ 1.65). When changing the Invasive z-score thresholds for grouping the TCGA samples, the most Invasive-high subgroup remains associated with the lowest survival (*Figure 5—figure supplement 2B,C*).

A further 158 samples that did not overlap with the RNA-seq set for which microarray expression data, survival information, and IDH status were available were used in a validation set. Individual samples were assigned to GSC clusters or C3 score classes as described above. The scores were then combined with those of the RNA-seq set for the survival analysis shown in *Figure 5—figure supplement 2D–E*.

## Acknowledgements

We would like to thank the Princess Margaret Genomics Centre for their assistance in this study and 10× Genomics, especially Adam Jerauld, Ariel Royall, and John Chevillet for training and support. We also thank Aude Gerbaud and Stacey Krunholtz for their help with figure formatting and David Vetrie for his thoughtful input on the manuscript.

## Additional information

### Funding

| Funder | Grant reference number | Author |
|---|---|---|
| Stand Up To Cancer Canada, Genome Canada, CIHR, OICR, AACR | SU2C-AACR-DT-19-15 | Michael D Taylor Samuel Weiss Peter B Dirks Mathieu Lupien |
| CIHR | TGH-158221 | Stephane Angers Peter B Dirks Mathieu Lupien |
| CIHR | MFE 338954 | Paul Guilhamon |
| Princess Margaret Cancer Foundation | | Mathieu Lupien |
| CIHR | | H Artee Luchman Samuel Weiss |
| Terry Fox Research Institute | | Stephane Angers Peter B Dirks |
| Hospital for Sick Children | | Peter B Dirks |
| Canadian Cancer Society | | Stephane Angers |
| NSERC | | Marco Gallo |
| Alliance for Cancer Gene | | Marco Gallo |

| Therapy | |
| --- | --- |
| Jessica's Footprint | Peter B Dirks |
| Hopeful Minds | Peter B Dirks |
| The Bresler Family | Peter B Dirks |

The funders had no role in study design, data collection and interpretation, or the decision to submit the work for publication.

### Author contributions

Paul Guilhamon, Conceptualization, Formal analysis, Funding acquisition, Validation, Investigation, Visualization, Writing - original draft, Writing - review and editing; Charles Chesnelong, Michelle M Kushida, Ana Nikolic, Divya Singhal, Christopher Arlidge, Naghmeh Rastegar, Xiaoguang Hao, Rozina Hassam, Heather Whetstone, Katrina I Ellestad, Investigation; Graham MacLeod, Nishani Rajakulendran, Jennifer Ai-wen Chan, Resources; Seyed Ali Madani Tonekaboni, Bettina Nadorp, Formal analysis; Florence MG Cavalli, Data curation; Laura J Smith, Investigation, Visualization; Fiona J Coutinho, Project administration; H Artee Luchman, Molly S Shoichet, Michael D Taylor, Benjamin Haibe-Kains, Samuel Weiss, Stephane Angers, Supervision; Marco Gallo, Supervision, Writing - review and editing; Peter B Dirks, Conceptualization, Supervision, Writing - original draft, Writing - review and editing; Mathieu Lupien, Conceptualization, Supervision, Funding acquisition, Writing - original draft, Project administration, Writing - review and editing

### Author ORCIDs

Paul Guilhamon https://orcid.org/0000-0001-8276-5987
Graham MacLeod http://orcid.org/0000-0001-6401-9307
Xiaoguang Hao https://orcid.org/0000-0003-2695-0111
Molly S Shoichet http://orcid.org/0000-0003-1830-3475
Stephane Angers http://orcid.org/0000-0001-7241-9044
Mathieu Lupien https://orcid.org/0000-0003-0929-9478

### Ethics

Human subjects: All tissue samples were obtained following informed consent from patients, and all experimental procedures were performed in accordance with the Research Ethics Board at The Hospital for Sick Children (Toronto, Canada), the University of Calgary Ethics Review Board, and the Health Research Ethics Board of Alberta - Cancer Committee (HREBA). Approval for access to pathological data was obtained from the respective institutional review boards.

Animal experimentation: All animal procedures were performed according to and approved by the Animal Care Committee of the Hospital for Sick Children or the University of Calgary. All attempts are made to minimize the handling time during surgery and treatment so as not to unduly stress the animals. Animals are observed daily after surgery to ensure there are no unexpected complications.

### Decision letter and Author response

Decision letter https://doi.org/10.7554/eLife.64090.sa1
Author response https://doi.org/10.7554/eLife.64090.sa2

## Additional files

### Supplementary files

- Supplementary file 1. List of GSC state labels, patient age, and sex.
- Supplementary file 2. Coordinates of discriminating ATAC regions for each GSC state.
- Transparent reporting form

## Data availability

The GSCs are available upon reasonable request from PBD and SW. The GSC ATAC-seq and DNA methylation data have been deposited at GEO (GSE109399). The scATAC-seq data has been deposited at GEO (GSE139136). RNA-seq data are available at EGA (EGAS00001003070).

The following datasets were generated:

| Author(s) | Year | Dataset title | Dataset URL | Database and Identifier |
|---|---|---|---|---|
| Guilhamon P, Kushida MM, Dirks PB, Lupien M | 2019 | Epigenetic characterization of glioblastoma stem cells | https://www.ncbi.nlm.nih.gov/geo/query/acc.cgi?acc=GSE109399 | NCBI Gene Expression Omnibus, GSE109399 |
| Nikolic A, Ellestad K, Singhal D, Gallo M | 2021 | Single-cell ATAC-Seq of Adult GBM | https://www.ncbi.nlm.nih.gov/geo/query/acc.cgi?acc=GSE139136 | NCBI Gene Expression Omnibus, GSE139136 |
| Guilhamon P, Kushida MM, Cavalli FMG, Dirks PB, Lupien M | 2018 | RNA-seq of Glioblastoma stem cells | https://ega-archive.org/search-results.php?query=EGAS00001003070 | EGA, EGAS00001003070 |

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
