## [Decision Letter]

**Acceptance summary:**

The current manuscript by Guilhamon et al. uses single cell sequencing technologies to identify and then characterize heterogeneities among glioblastoma (GBM) stem cells (GSC), which extend beyond current disease classifications. This is the first study to apply single cell sequencing specifically to GSCs with the goal of identifying GSC subtypes using primarily chromatin accessibility signatures. Moreover, the use of chromatin accessibility signatures in combination with TF motif prediction to guide identification of transcriptional dependencies is novel in the context of GSC studies.

**Decision letter after peer review:**

[Editors’ note: the authors submitted for reconsideration following the decision after peer review. What follows is the decision letter after the first round of review.]

Thank you for submitting your work entitled "Single-cell chromatin accessibility in glioblastoma delineates cancer stem cell heterogeneity predictive of survival" for consideration by *eLife*. Your article has been reviewed by three peer reviewers, one of whom is a member of our Board of Reviewing Editors, and the evaluation has been overseen by a Senior Editor. The following individual involved in review of your submission has agreed to reveal their identity: Roel Verhaak (Reviewer #3).

Our decision has been reached after consultation between the reviewers. Based on these discussions and the individual reviews below, we regret to inform you that your work will not be considered further for publication in *eLife*, due to the need for revisions that would likely take longer than 2-3 months. We would however encourage resubmission once the revisions are completed.

Summary

The current manuscript by Guilhamon et al. uses single cell sequencing technologies to identify and then characterize heterogeneities among glioblastoma (GBM) stem cells (GSC), which extend beyond current disease classifications. The paper was reviewed by three experts, who all agreed that the work will add meaningful knowledge to the field; but that more experimentation would be needed to better support the claims as made. The reviewers agreed that this is the first study to apply single cell sequencing specifically to GSCs with the goal of identifying GSC subtypes using primarily chromatin accessibility signatures. In particular, the use of chromatin accessibility signatures in combination with TF motif prediction to guide identification of transcriptional dependencies is novel in the context of GSC studies. However, it was felt that a number of enhancements would be needed, including greater validation of the cell samples, the inclusion of additional samples, and clarification of some of the methodologies employed. We encourage resubmission once these issues have been addressed.

Major concerns:

1) The tumor cell and GSC population enrichment should be validated using functional and genomic methods (including evidence for either Chr7 amplification or Chr10 deletion), and additional patient samples should be analyzed, preferably without extended culture times. Indeed, there appeared to be a low correlation between GBM stem TF enrichment and GSC signature enrichment and the degree of concordance between samples was not fully developed. It would be important to have more samples for many reasons, including the idea presented by the authors of differences between tumors in the dominant cell type. GBM has significant variation spatially, as well, so having only 4 tumors seems underpowered.

2) Details about the tumor collection should be included as should full information about tumor genetics, patient information, and treatment for each sample.

3) Figure 1 should be revised to clarify cells which fail to classify and to signify whether shared chromatin states exist across patients.

4) Figure 3 should be revised. Data pertaining to GSCs should be compared to the published GBM scRNAseq data to check the subtypes (Neftel et al., 2019, etc.). In addition, the top and bottom panels could be combined, and the sample assignment of TCGA subtypes, chromatin clusters and expression clusters should be better presented in the sankey diagram. The GSC samples and color bars showing chromatin clusters should be kept consistent between the top and the bottom panels. Figure 3A is missing a legend as there is no indication of what the intensity values represent. For Figure 3B, please clearly indicate an associated P-value testing for the enrichment of TCGA subtypes among the GSC ATAC states. Figure 3C should present cut-offs for what was considered statistically significant (i.e., a dotted line). Importantly, "Figure 3—figure supplement 1" presents the most differentially accessible regions between GSC states. Notably, ChrX is present in all. Chromatin accessibility read-outs might be significantly impacted by the presence of an inactive X chromosome. Hence an imbalance in the sex distribution in the GSC states might pick up sex-specific and not tumor-specific differences. Thus, the authors should show that these regions do not vary with patient sex. Finally, the evidence that "GSC states are not genetically defined (Figure 3D)" is derived from DNA methylation copy number, and a signature of selected genes. Bona fide GBM genes such as PDGFRA, NF1, AKT3, QKI should also be included in this analysis and the possibility that the GSC states cluster by frequent chromosome arm-level events (chr7, chr19, chr20 gains and chr10 loss) should be entertained. If they still do not cluster by GSC state, then it is perhaps more accurate to say that "GSC states are not defined by somatic copy number events" since there is no gene mutation data presented for these samples.

5) The validation of the states in the TCGA data are not strong. The cut-offs seem arbitrary and most of the samples seem to fall into a single group. Is this due to sampling bias? There is no connection to tumor genetics. Also, the bulk GBM ATACseq tumor profiles were generated in the TCGA datasets that presumably have accompanying RNA, DNA, and methylation data (Corces et al., Science 2018). While this represents a small dataset, it would be advantageous to incorporate analysis of these more complex tissue samples alongside the already presented data to provide confidence in generalizability. The concordance between RNA-seq and ATAC-seq data for the paired datasets should also be presented.

6) Invasion assays should include all subtypes in order to highlight properties of the invasive subtype.

7) In order to support the claim that the "Invasive-high" group confers a worse prognosis, a multivariate analysis should be employed if possible.

8) There are a number of clerical errors that should be addressed:

– Figure 4—figure supplement 3A is blank

– Please double check all sample names, in Figure 3B, G613 changes from invasive to reactive and there are several instances where the sample names do not match up.

– If other samples (outside of the 27 discussed in Figure 3) were used in Figure 4, it should be made clear. G472, G440, and G473 are not in Figure 3

– A table with all the sample names, GSC subtype identified through ATAC-seq, and validation of GSC should be included

– Figure 4—figure supplement 1 G is missing labels

– Please include the GSEA gene set size.

– Describe how many essentiality genes were included for GSEA genes

– Figure 4D should be revised to better represent the number of mice were used. A detailed experimental design and the survival curve analysis should also be provided.

– Figure 2—figure supplement 1 lacks sample annotation in both the image and the figure legend.

– Figure 4—figure supplement 3 was blacked out in the version provided for review.

– There appears to be a typographical error at the start of Results paragraph five, "FOXD1 is essential only in the Reactive state […]". This seems like it should be, "Invasive".

Reviewer #1:

The current manuscript by Guilhamon et al. uses single cell sequencing technologies to identify and then characterize heterogeneities among glioblastoma (GBM) stem cells (GSC), which extend beyond current disease classifications. Single cell technologies have already been applied to GBM in various publications. The identification of an invasive GSC subtype associated with poor survival is not novel, as it largely complements the previously identified mesenchymal GSC subtype. However, as far as I'm aware this is the first study to apply single cell sequencing specifically to glioblastoma stem cells (GSCs) with the goal of identifying GSC subtypes using primarily chromatin accessibility signatures. In particular, the use of chromatin accessibility signatures in combination with TF motif prediction to guide identification of transcriptional dependencies is novel in the context of GSC studies. Thus, this manuscript would constitute a meaningful contribution to the field of GBM and epigenetics.

– Validation of GSC population enrichment should be done and shown. Particularly because despite what is described in the text, Figure 2F show low correlation between GBM stem TF enrichment and GSC signature enrichment. 2 out of 4 primary GBM show very little correlation. Therefore, GSC enrichment should be validated with other methods, such as staining for known neural stem cell markers or clustering/enrichment/scoring based on validated GSC gene sets.

– Chromatin accessibility scores at promoter regions is used as a surrogate for gene expression for numerous analyses. What is the concordance between RNA-seq and ATAC-seq for the paired datasets? Particularly, what is the concordance for the subtype specific peaks? The concordance of gene expression and chromatin accessibility feels particularly important when exclusive peaks only account for ~5-15% of all peaks.

– Invasion assays should include all subtypes in order to highlight properties of the invasive subtype. Additionally, the author's conclusions would be strengthened by the combination of FOXD1 knockdown/inhibition with the invasion assays.

– Figure 4—figure supplement 3A is blank.

– Please double check all sample names, in Figure 3B, G613 changes from invasive to reactive. There are several instances where the sample names do not match up.

– If other samples (outside of the 27 discussed in Figure 3) were used in Figure 4, it should be made clear. G472, G440, and G473 are not in Figure 3.

– Please include table with all the sample names, GSC subtype identified through ATAC-seq, and validation of GSC.

– Figure 4—figure supplement 1 G is missing labels.

Reviewer #2:

In this manuscript, Guilhamon et al. describe a single cell ATACseq study in GBM patient tissues and bulk ATACseq in a panel of patient-derived GBM stem cell cultures. Using this dataset, the authors studied the three states in glioblastoma stem cell cultures, which presents some interesting findings to the biology of GBM. The dataset generated in this study is useful to the field. There are some concerns to be addressed:

The single cell data would seem to be potentially the most important, but the number of samples is small and the degree of concordance between samples was not fully developed. It would be important to have more samples for many reasons, including the idea presented by the authors of differences between tumors in the dominant cell type. GBM has significant variation spatially, as well, so having only 4 tumors seems underpowered.

There are missing details about the tumor collection. The three groups are designated to include reactive, which has immune cell signatures, and invasive. Often the tumor does not include areas of necrosis and invasion in most surgical specimens.

Why were the GSC cultured for 5-12 passages? This would seem to induce potential selection or change the cells. If the laminin conditions are optimal, why weren't the GSC studied immediately? It would be helpful to include support for the LDAs and xenograft studies for each of the lines.

Please include the full information about tumor genetics, patient information, and treatment for each sample.

There are many very strong statements made. It would seem premature to call single cells GSC only based on transcription. I am not aware that any profile is validated to claim absolutely to identify GSC. This is but one of the leaps that were made. I would ask the authors to be more careful and simply identify exactly what was measured.

There is no direct comparison of GBM tissues and GSC cultures. The conclusion that GSCs "reflect the chromatin identity of putative cancer stem cells found in primary brain tumors" should be supported by more evidence. For examples, direct comparison of some important drivers or markers at the chromatin level should be provided.

The analysis of GBM cancer stem cell signature and 4 cellular states is interesting. Can the authors explain why the putative cancer stem cells were under-represented in MES cell state? Did the authors consider or separate transcription factors or markers for cancer stem cells of different types in GBM? Is there any difference between stem and non-stem cells from the same cellular state?

Is there overlap between 4 GBM tissues with scATAC data and patients with GSC culture?

The description of the mismatch between chromatin and TCGA subtype is confusing (Figure 3B). According to the description, GSCs are believed to be a relatively pure population, can the authors compare GSCs to the published GBM scRNAseq data to check the subtypes (Neftel et al., 2019, etc.)? In addition, the top and bottom panels could be combined, and the sample assignment of TCGA subtypes, chromatin clusters and expression clusters should be better presented in the sankey plot. The GSC samples and color bars showing chromatin clusters are not consistent between the top and the bottom panels, making it really difficult to understand.

It's interesting that many GSCs express immune cell related genes, which should be further explained, as the enrichment significance (FDR) is not very strong.

The differentially expressed gene analysis between three GSC states should be performed to provide state markers to have a better understanding of cellular states.

Figure 4D is difficult to understand. What's the meaning of the n? How many mice were used in the experiment? A detailed experimental design and the survival curve analysis should be provided.

The CRISPR screen would be beneficial to validate, including additional direct studies of specific targets in a larger cohort of the GSC.

The validation of the states in the TCGA data are not strong. The cutoffs seem arbitrary and most of the samples seem to fall into a single group. Is this due to sampling bias? There is no connection to tumor genetics.

Reviewer #3:

Guilhamon et al. performed single-cell ATACseq in four primary IDH-wildtype glioblastomas to derive chromatin accessibility signatures of heterogeneous glioma stem cell populations. The scATACseq primary tumor analysis suggested the existence of three major modules: Reactive, Constructive, and Invasive that define glioma stem populations and are present to varying degrees within a single tumor. The module/cell state relevance is supported by leveraging new and existing functional assay data in sets of patient-derived GSC populations. The authors then demonstrate that transcription factors nominated by DNA recognition motif enrichment in each GSC state are preferentially essential through analysis of a recently published genome-wide CRISPR/Cas9 essentiality screen. Finally, evidence is presented to suggest that these chromatin signatures are associated with survival in a xenograft model and TCGA subjects.

Overall, this study addresses an important biological question of glioblastoma stem cell heterogeneity and is one of the earliest papers investigating scATAC glioblastoma profiles. However, the conclusions drawn are not fully supported by the results and in some cases the technical details of the experiment/analysis are lacking. The following suggestions are intended to help strengthen this work.

1) The clarity of the paper would be aided by additional technical descriptions throughout the manuscript. For example:

a) The four GBM samples profiled using scATACseq were classified using the Neftel signatures, on the basis of accessibility of signature gene promoters. This approach will bin individual cells whether the classification is accurate, or not. How can we be sure that this approach accurately provides Neftel classification?

b) The same classification method is suggested to result in greater stratification of the Neftel MES category. This result may be confounded by the presence of tumor-associated microglia that will be classified as mesenchymal. Were all classified cells tumor cells and how was this validated?

c) It was not fully explained why single cells without evidence for either Chr7 amplification or Chr10 deletion were included in further analyses regarding glioma stem cell populations. The cells that were not labelled by "GBM CNV+" (Figure 2—figure supplement 1) also seem to less frequently be classified as "STEM" (Figure 1D, especially G4250 and G4275). Are these cells of poorer quality or do they represent non-tumor cell types? It would also be helpful to know whether the authors enriched for tumor cells during the dissociation process (otherwise one would expect non-tumor cells too), the number of nuclei that were loaded for 10X, basic quality control metrics for each sample, and if cells identified as non-tumor were excluded from construction of the chromatin modules.

2) Figure 1 was challenging to follow with the multiple annotations of the patient UMAPs. For instance, in panels 1B and 1D the original numbered clusters remain for some not all cells. Are these cells that failed to confidently classify as cell states (i.e., AC, MES, NPC, OPC) or stem states? Perhaps consider "graying" out cells that fail to classify and explain why not all tumor cells might have a cell state classification would benefit the audience. I could not find mention of this in the Results or Materials and methods sections. Also, what happens when all cells from the four subjects are analyzed together in a single analysis? Do shared chromatin states that exist across patients emerge or is there highly specific patient clustering that reflects inter-patient heterogeneity? Panel C in this figure was also difficult to understand. None of the modules equal to 100%, is this because some cells failed to classify?

3) Bulk GBM ATACseq tumor profiles were generated in the TCGA dataset that presumably have accompanying RNA, DNA, and methylation data (Corces et al. Science 2018). While this represents a small dataset, it would be advantageous to incorporate analysis of these more complex tissue samples alongside the already presented data to provide confidence in generalizability.

4) Figure 3 appears to be missing some critical information. Figure 3A is missing a legend as there is no indication what the intensity values represent. For Figure 3B, is there an associated P-value testing for the enrichment of TCGA subtypes among the GSC ATAC states? Figure 3C could present cut-offs for what was considered statistically significant (i.e., a dotted line). Importantly, "Figure 3—figure supplement 1" presents the most differentially accessible regions between GSC states. While it is difficult to read the all the regions it's notable that ChrX is present in all. I imagine that chromatin accessibility read-outs might be significantly impacted by the presence of an inactive X chromosome. If there is an imbalance in the sex distribution in the GSC states, then this might skew to picking up sex-specific and not tumor-specific differences. Have the authors shown that these regions do not vary with patient sex.

5) The set of GSCs making up the Invasive subtype are all classified as Mesenchymal by gene expression (Figure 3B), consider calling it as such for consistency?

6) The evidence that "GSC states are not genetically defined (Figure 3D)" is derived from DNA methylation copy number, and a signature of selected genes. Is this sufficient? What about bona fide glioblastoma genes such as PDGFRA, NF1, AKT3, QKI? Did DNA copy number profiles align with transcriptional subtypes? Do the GSC states cluster by frequent chromosome arm-level events (chr7, chr19, chr20 gains and chr10 loss)? If they still do not cluster by GSC state, then it is perhaps more accurate to say that "GSC states are not defined by somatic copy number events" since there is no gene mutation data presented for these samples.

7) More than one publication has speculated to be able to stratify IDH wild type glioblastoma into prognostically relevant groups, i.e. PMID 30753603. For a tumor type with a single treatment that is not very effective, it is unclear that predicting prognosis has relevance.

8) In a multi-variate model that includes known prognostic factors (i.e., subject age and MGMT promoter methylation status) does the "Invasive-high" group still confer a worse prognosis? This would aid the author's claim that the signature is "predictive of low patient survival" independent on known prognostic factors.

9) Single-nucleus ATACseq of IDH mutant glioma has been reported (PMID 31806013) and there is some overlap in the gene sets that were identified to associate with cell states and survival, and it would be appropriate to cite this literature.

[Editors’ note: further revisions were suggested prior to acceptance, as described below.]

Thank you for resubmitting your work entitled "Single-cell chromatin accessibility of glioblastoma identifies invasive cancer stem cell linked to lower survival" for further consideration by *eLife*. Your revised article has been evaluated by Kevin Struhl (Senior Editor) and a Reviewing Editor.

The manuscript has been improved but there are some remaining issues that need to be addressed before acceptance, as outlined below:

The current manuscript by Guilhamon et al. uses single cell sequencing technologies to identify and then characterize heterogeneities among glioblastoma (GBM) stem cells (GSC), which extend beyond current disease classifications. The paper will add meaningful knowledge to the field and revisions have enhanced this manuscript. For example, validation of the cells was provided, methodologies were clarified, and impressive data related to the invasive subtype was provided. Additional samples would have enhanced the study, but were difficult to obtain given the COVID-19 pandemic. Overall, though, this would be the first study to apply single cell sequencing specifically to GSCs with the goal of identifying GSC subtypes using primarily chromatin accessibility signatures. Moreover, the use of chromatin accessibility signatures in combination with TF motif prediction to guide identification of transcriptional dependencies is novel in the context of GSC studies. These strengths overcome weaknesses associated with small sample numbers.

1) Please clarify what percentage of cells were estimated to be stem cells, among the scATACseq datasets.

2) Please modify the statement "A more likely possibility is that given the TCGA subtypes were determined from bulk GBM, they may not fully capture the nature of rarer populations found within a tumor, such as the cancer stem cell populations." Due to the amount of testing needed to test this, please modify this explanation to be broader and to also include the possibility that this is due at least in part to differences in microenvironments.

3) Please modify the statement "No molecular signature in GBM has so far been reported that can significantly stratify the poorer prognosis IDH wild-type patients by survival." This is incorrect. DNA methylation profiling classifies IDH wild type tumors into several categories including Mesenchymal-like (Ceccarelli et al., Cell, 2016), which shows significantly worse outcomes even among only MGMT-unmethylated tumors.

4) Regarding the statement "Using the TCGA IDH wild type cohort (n=144)", please clarify that: a) this cohort relates to the subset of GBM IDH wildtype cases with available RNAseq data (presumably); and b) that the GSC states were converted from a chromatin accessibility signature to a RNA expression signature.

5) There is no obvious reason why the TCGA IDH wildtype cohort could not also be classified on the basis of the available Affymetrix microarray data, which would enable analysis of a substantially larger cohort in pursuit of a survival difference (n=400). Please include this analysis.

6) Please include a small paragraph in the Discussion about the limitations of the study; particularly as they relate to a potentially small samples size.

Reviewer #3:

A few remaining comments with respect to the revision:

1) It would be helpful to clarify what percentage of cells were estimated to be stem cells, among the scATACseq datasets.

2) "A more likely possibility is that given the TCGA subtypes were determined from bulk GBM, they may not fully capture the nature of rarer populations found within a tumor, such as the cancer stem cell populations." While in this reviewer's opinion the most likely explanation is the presence/absence of a tumor microenvironment, we won’t know without a substantial amount of extra testing, which is beyond the scope. Please modify as to keep the explanations broad.

3) "No molecular signature in GBM has so far been reported that can significantly stratify the poorer prognosis IDH wild-type patients by survival." This is incorrect. DNA methylation profiling classifies IDH wild type tumors into several categories including Mesenchymal-like (Ceccarelli et al., Cell, 2016), which shows significantly worse outcomes even among only MGMT-unmethylated tumors.

4) "Using the TCGA IDH wild type cohort (n=144)" it would help to clarify that a. this cohort relates to the subset of GBM IDH wildtype cases with available RNAseq data (presumably); and b. that the GSC states were converted from a chromatin accessibility signature to a RNA expression signature.

5) There is no obvious reason why the TCGA IDH wildtype cohort could not also be classified on the basis of the available Affymetrix microarray data, which would enable analysis of a substantially larger cohort in pursuit of a survival difference (n=400).

---

## [Author Response]

[Editors’ note: the authors resubmitted a revised version of the paper for consideration. What follows is the authors’ response to the first round of review.]

Major concerns:1) The tumor cell and GSC population enrichment should be validated using functional and genomic methods (including evidence for either Chr7 amplification or Chr10 deletion), and additional patient samples should be analyzed, preferably without extended culture times. Indeed, there appeared to be a low correlation between GBM stem TF enrichment and GSC signature enrichment and the degree of concordance between samples was not fully developed. It would be important to have more samples for many reasons, including the idea presented by the authors of differences between tumors in the dominant cell type. GBM has significant variation spatially, as well, so having only 4 tumors seems underpowered.

Evidence of chr7 and chr10 alterations at the single cell level can be found in Figure 2—figure supplement 1A-D, with details presented in the Materials and methods section “Single cell ATAC-seq”, and in the main text. To further explore the concordance between samples in response to this and other comments below, we also annotated single cells that failed to classify into one of the Neftel et al. subtypes, and the number of tumours sharing UMAP modules of each state. These data can be found in Figure 1—figure supplement 1A-B, and in the main text.

We also conducted additional analyses into the concordance of stem cell calls by the two signatures (GSCs and the 19 stem TF signature by Suva et al., 2014). We found that the overlap in cells significantly enriched for either stem signature across the four tumours was significant (hypergeometric test p-value = 8.8 x 10^−8^), and this has been added to the main text.

Restrictions due to the covid-19 pandemic prevented us from obtaining and processing new samples for scATAC-seq. We however attempted to incorporate new samples into our single-cell analysis using publicly available data from the Wang et al. (Cancer Discovery, 2019) study. That study contained four IDH-wt GBM samples processed for scATAC-seq, of which three had raw data available (one sample’s data was corrupted on the EGA archive). Unfortunately, only one of the three samples had more than 350 cells available after quality control, and that sample had only 19% sequencing saturation (as compared to ~75% in our samples). Under these circumstances it was unfeasible to integrate these data into our study and perform on them the same analyses used on our samples.

Collectively, the data presented in this manuscript represent the most comprehensive study to date of single cell chromatin accessibility in glioblastoma and, as highlighted in your summary above, provide novel insights and a valuable resource to the community.

2) Details about the tumor collection should be included as should full information about tumor genetics, patient information, and treatment for each sample.

We have included details of tumor collection for those samples used in the scATAC-seq as well as known patient characteristics (IDH status, sex, and age) in the Materials and methods section “Single cell ATAC-seq”. Details on sample collection for the GSCs are included in the Materials and methods section “Patient samples and cell culture”. Details on the GSC state of each GSC are in Supplementary Table 1.

3) Figure 1 should be revised to clarify cells which fail to classify and to signify whether shared chromatin states exist across patients.

We have added that information in the form of stacked bar plots in Figure 1—figure supplement 1A.

4) Figure 3 should be revised. Data pertaining to GSCs should be compared to the published GBM scRNAseq data to check the subtypes (Neftel et al., 2019, etc.). In addition, the top and bottom panels could be combined, and the sample assignment of TCGA subtypes, chromatin clusters and expression clusters should be better presented in the sankey diagram. The GSC samples and color bars showing chromatin clusters should be kept consistent between the top and the bottom panels. Figure 3A is missing a legend as there is no indication of what the intensity values represent. For Figure 3B, please clearly indicate an associated P-value testing for the enrichment of TCGA subtypes among the GSC ATAC states. Figure 3C should present cut-offs for what was considered statistically significant (i.e., a dotted line). Importantly, "Figure 3—figure supplement 1" presents the most differentially accessible regions between GSC states. Notably, ChrX is present in all. Chromatin accessibility read-outs might be significantly impacted by the presence of an inactive X chromosome. Hence an imbalance in the sex distribution in the GSC states might pick up sex-specific and not tumor-specific differences. Thus, the authors should show that these regions do not vary with patient sex. Finally, the evidence that "GSC states are not genetically defined (Figure 3D)" is derived from DNA methylation copy number, and a signature of selected genes. Bona fide GBM genes such as PDGFRA, NF1, AKT3, QKI should also be included in this analysis and the possibility that the GSC states cluster by frequent chromosome arm-level events (chr7, chr19, chr20 gains and chr10 loss) should be entertained. If they still do not cluster by GSC state, then it is perhaps more accurate to say that "GSC states are not defined by somatic copy number events" since there is no gene mutation data presented for these samples.

Following the reviewer’s recommendations we have combined the top and bottom panels of Figure 3B, and corrected the sample labelling on the heatmap. We thank the reviewers for pointing out that error. We have additionally added a legend to both heatmaps on Figure 3A and 3B and added the GSEA Enrichment Score range to Figure 3B.

In Figure 3C only significant (q-value <=0.05 after multiple test correction) terms are displayed. We have included this in the revised figure legend.

The reviewers correctly point out that sex imbalance between groups could have been a concern; we had verified there was none early in the study but failed to discuss it in the original version of this manuscript. This is now included in the Materials and methods section “ATAC-seq”.

On the reviewers’ recommendation, we have extensively updated Figure 3D, including additional samples, as well as more GBM-related genes and the large genomic events most commonly found in GBM, including on chromosomes 7,10,19 and 20. The conclusions from this analysis are maintained and we have amended the text as suggested.

5) The validation of the states in the TCGA data are not strong. The cut-offs seem arbitrary and most of the samples seem to fall into a single group. Is this due to sampling bias? There is no connection to tumor genetics. Also, the bulk GBM ATACseq tumor profiles were generated in the TCGA datasets that presumably have accompanying RNA, DNA, and methylation data (Corces et al. Science 2018). While this represents a small dataset, it would be advantageous to incorporate analysis of these more complex tissue samples alongside the already presented data to provide confidence in generalizability. The concordance between RNA-seq and ATAC-seq data for the paired datasets should also be presented.

The cut-offs and methods used in the TCGA state labelling are detailed in the Materials and methods section “Survival Analysis”. They were set at z-score of 1.65 (which corresponds to a p-value of 0.05) for the Invasive-high to identify those patients with a significant enrichment of the Invasive GSC signature. For the Invasive-mid patients, we chose a z-score between 1 and 1.65 in order to identify those patients who displayed an important enrichment of the signature even though it did not reach significance. In addition, we show in Figure 5—figure supplement 2B-C, that changing these thresholds does not alter the conclusions of the analysis.

While we appreciate the suggestion to use the Corces et al. datasets to match expression and chromatin accessibility, the 9 GBM ATAC-seq samples used in that study do not have RNA-seq data available. This can be verified in the supplementary table of the study as well as directly on the GDC data portal.

6) Invasion assays should include all subtypes in order to highlight properties of the invasive subtype.

We have performed numerous additional invasion assays to confirm the invasive properties of the invasive GSC state. Using a minimum of three representative GSC lines from each of the three GSC states as well as two human fetal neural stem cells (hfNSCs) as controls, we performed collagen invasion assays. For each GSC, the assay was repeated in triplicate, and at least 3 spheres were measured for each replicate. The results confirm that Invasive state GSCs are over 3-fold more invasive than GSCs from the other states or the control hfNSCs. These data can be found in Figure 5 B-C, Figure 5—figure supplement 1, in the Materials and methods section “Invasion assays”, and in the main text.

7) In order to support the claim that the "Invasive-high" group confers a worse prognosis, a multivariate analysis should be employed if possible.

With this analysis, we were investigating whether a particular GSC subtype enriched in a given tumour might be associated with survival. To do this we calculated the hazard ratios of the different GSC states both in the xenografts and TCGA data as shown in the Results. As no other molecular signature in GBM has so far been reported that can significantly stratify the poorer prognosis IDH wild-type patients by survival, and as using profiles from GSCs specifically has not been attempted before, there are no known covariates we could integrate into this analysis.

8) There are a number of clerical errors that should be addressed:– Figure 4—figure supplement 3A is blank– Please double check all sample names, in Figure 3B, G613 changes from invasive to reactive and there are several instances where the sample names do not match up.– If other samples (outside of the 27 discussed in Figure 3) were used in Figure 4, it should be made clear. G472, G440, and G473 are not in Figure 3– A table with all the sample names, GSC subtype identified through ATAC-seq, and validation of GSC should be included– Figure 4—figure supplement 1 G is missing labels– Please include the GSEA gene set size.– Describe how many essentiality genes were included for GSEA genes– Figure 4D should be revised to better represent the number of mice were used. A detailed experimental design and the survival curve analysis should also be provided.– Figure 2—figure supplement 1 lacks sample annotation in both the image and the figure legend.– Figure 4—figure supplement 3 was blacked out in the version provided for review.– There appears to be a typographical error at the start of Results paragraph five, "FOXD1 is essential only in the Reactive state […]". This seems like it should be, "Invasive".

We are grateful to the reviewers for pointing out these errors. We have corrected them as follows:

– The invasion data is no longer in Figure 4—figure supplement 3A but in Figure 5 and Figure 5—figure supplement 1.

– The labels in Figure 3B have been corrected

– Additional samples used beyond the 27 discussed in Figures 2 and 3 are outlined in the Materials and methods section “ATAC-seq”

– A table with sample names and GSC states has been added as Supplementary file 1

– Labels have been added to Figure 4—figure supplement 1G

– The GSEA gene set sizes have been added to the figure legend

– Essentiality gene numbers used in the GSEA analysis have been added to the Materials and methods section “Gene Essentiality Screen”

– We have added the number of mice used for each of the 37 GSCs in the Materials and methods section “Survival Analysis” and in the figure legend. An average of 5 mice were used for each GSC.

– Annotation has been added to Figure 2—figure supplement 1 in the image and figure

– As mentioned above, the invasion data is no longer in Figure 4 but in Figure 5 and Figure 5—figure supplement 1.

– The typographical error related to FOXD1 has been corrected in the main text.

[Editors’ note: what follows is the authors’ response to the second round of review.]

The current manuscript by Guilhamon et al. uses single cell sequencing technologies to identify and then characterize heterogeneities among glioblastoma (GBM) stem cells (GSC), which extend beyond current disease classifications. The paper will add meaningful knowledge to the field and revisions have enhanced this manuscript. For example, validation of the cells was provided, methodologies were clarified, and impressive data related to the invasive subtype was provided. Additional samples would have enhanced the study, but were difficult to obtain given the COVID-19 pandemic. Overall, though, this would be the first study to apply single cell sequencing specifically to GSCs with the goal of identifying GSC subtypes using primarily chromatin accessibility signatures. Moreover, the use of chromatin accessibility signatures in combination with TF motif prediction to guide identification of transcriptional dependencies is novel in the context of GSC studies. These strengths overcome weaknesses associated with small sample numbers.1) Please clarify what percentage of cells were estimated to be stem cells, among the scATACseq datasets.

This information has now been added within the Results section as well as within the Materials and methods (Single Cell Atac section).

2) Please modify the statement "A more likely possibility is that given the TCGA subtypes were determined from bulk GBM, they may not fully capture the nature of rarer populations found within a tumor, such as the cancer stem cell populations." Due to the amount of testing needed to test this, please modify this explanation to be broader and to also include the possibility that this is due at least in part to differences in microenvironments.

We have now broadened the statement in the main text and included the possibility of the absence of a tumor microenvironment in the GSC populations. This change can be found in the Results section.

3) Please modify the statement "No molecular signature in GBM has so far been reported that can significantly stratify the poorer prognosis IDH wild-type patients by survival." This is incorrect. DNA methylation profiling classifies IDH wild type tumors into several categories including Mesenchymal-like (Ceccarelli et al., Cell, 2016), which shows significantly worse outcomes even among only MGMT-unmethylated tumors.

In our own review of the manuscript cited by the reviewer, we could not find data that directly supports the identification of a subset of adult IDH-wt GBM tumors with a worse prognosis, particularly as the cohorts used mixed both low-grade gliomas and GBM as well as pediatric and adult samples. However, in order to clarify the statement in the manuscript, we have modified it to reflect that this claim applies only to IDH-wt adult GBM and phrased as follows “molecular signature in adult GBM stratifying the poorer prognosis *IDH* wild-type patients by survival are lacking.”

4) Regarding the statement "Using the TCGA IDH wild type cohort (n=144)", please clarify that: a) this cohort relates to the subset of GBM IDH wildtype cases with available RNAseq data (presumably); and b) that the GSC states were converted from a chromatin accessibility signature to a RNA expression signature.

We have expanded the Materials and methods section related to the Survival Analysis to include these clarifications and specify how the TCGA cohort was identified as well as the full details of the conversion between the chromatin accessibility and RNA expression signatures.

5) There is no obvious reason why the TCGA IDH wildtype cohort could not also be classified on the basis of the available Affymetrix microarray data, which would enable analysis of a substantially larger cohort in pursuit of a survival difference (n=400). Please include this analysis.

We thank the reviewer for this suggestion. Incorporating the microarray expression data confirms our findings. We have included the results of this analysis in the following text and figures: Results , Figure 5—figure supplement 2D-E and Materials and methods survival analysis section.

Of note, the results of the survival analysis obtained using the TCGA microarray data are very similar to those obtained using the RNAseq (p-value=0.017 vs 0.019) and are featured in the supplementary figure as validation. However, as only 54% of our signature genes were included in the microarray dataset, the original classification based on RNAseq remains more robust and we have elected to keep that original classification within the main manuscript figure.

6) Please include a small paragraph in the Discussion about the limitations of the study; particularly as they relate to a potentially small samples size.

We have now included a paragraph discussion this limitation in the Discussion section.